# Dietary AhR Ligands Have No Anti-Fibrotic Properties in TGF-β1-Stimulated Human Colonic Fibroblasts

**DOI:** 10.3390/nu14163253

**Published:** 2022-08-09

**Authors:** Asma Amamou, Linda Yaker, Mathilde Leboutte, Christine Bôle-Feysot, Guillaume Savoye, Rachel Marion-Letellier

**Affiliations:** 1INSERM Unit 1073, University of Rouen, CEDEX, 76183 Rouen, France; 2Institute for Research and Innovation in Biomedicine (IRIB), University of Rouen, CEDEX, 76183 Rouen, France; 3Gastroenterology Department, Rouen University Hospital, CEDEX, 76031 Rouen, France

**Keywords:** aryl hydrocarbon receptor, CCD-18Co, curcumin, intestinal fibrosis, tryptophan derivatives

## Abstract

Background: Intestinal fibrosis is a common complication in inflammatory bowel disease (IBD) patients without specific treatment. Aryl hydrocarbon receptor (AhR) activation is associated with better outcomes in intestinal inflammation. Development of novel therapies targeting fibrogenic pathways is required and we aimed to screen dietary AhR ligands for their anti-fibrotic properties in TGF-β1-stimulated human colonic fibroblast cells. Methods: The study was conducted using TGF-β1-stimulated CCD-18Co, a human colonic fibroblast cell line in response to increased concentrations of dietary ligands of AhR such as FICZ, ITE, L-kynurenine and curcumin. Fibrosis markers such as α-SMA, COL1A1, COL3A1 and CTGF were assessed. AhR and ANRT RNA were evaluated. Results: TGF-β1 at 10 ng/mL significantly induced mRNA levels for ECM-associated proteins such as CTGF, COL1A1 and COL3A1 in CCD-18Co cells. FICZ from 10 to 1000 nM, L-kynurenine from 0.1 to 10 μM, ITE from 1 to 100 μM or curcumin from 5 to 20 μM had no significant effect on fibrosis markers in TGF-β1-induced CCD-18Co. Conclusions: Our data highlight that none of the tested dietary AhR ligands had an effect on fibrosis markers in TGF-β1-stimulated human colonic fibroblast cells in our experimental conditions. Further studies are now required to identify novel potential targets in intestinal fibrosis.

## 1. Introduction

Intestinal fibrosis is a major complication of inflammatory bowel disease (IBD) without a specific therapy [1]. Myofibroblast activation is crucial in intestinal fibrogenesis. These cells are derived from various cell origins. Upon activation, myofibroblasts produce extracellular matrix (ECM) proteins to promote fibrogenesis instead of normal healing [1]. Intestinal fibrosis is thus characterized by excessive ECM accumulation by activated myofibroblasts resulting in increased tissue stiffness and progressive functional damage [1]. Over the last decade, aryl hydrocarbon receptor (AhR) has been positively associated with anti-fibrotic molecular targets in fibrosis from other organs [2]. As the mechanisms behind fibrogenesis in the gut are believed to be similar to those from extra-intestinal organs, it may be relevant to investigate whether AhR activation leads to intestinal fibrosis inhibition.

AhR is a member of the basic helix–loop–helix–(bHLH) superfamily of transcription factors, which are first associated with cellular responses to xenobiotics [3,4]. More recently, dietary ligands such as tryptophan metabolites or curcumin have also been described as AhR ligands [3,4]. AhR is highly expressed in the gastro-intestinal tract and its activation has been associated with beneficial effects in the gut [5]. Administration of 6-formylindolo(3,2-b) carbazole (FICZ), an AhR agonist, regulated colon inflammation in -mice deleted in Card9 [6]. Very interestingly, the authors of this study also found that gut microbiota from IBD patients exhibited a reduced production of AhR ligands [6]. Nutrients are able to AhR and regulate inflammatory responses. FICZ also exhibit anti-fibrotic properties in the intestine [7]. From other fibrotic diseases, it has been shown that dietary ligands of AhR such as 2-(1′H-indole-3′-carbonyl)-thiazole-4-carboxylic acid methyl ester (ITE), L-kynurenine [8] or curcumin [9,10] can inhibit ECM-associated proteins in fibroblasts from various organs.

We thus aimed to evaluate whether dietary AhR agonists may reduce intestinal fibrogenesis through activation of myofibroblasts. 

## 2. Materials and Methods

### 2.1. In Vitro Model Reagents

Human recombinant TGF-β1 was obtained from PeproTech, (Cranbury, NJ, USA). FICZ and kynurenine were purchased from Sigma (St. Louis, MO, USA). ITE and curcumin were purchased from TOCRIS (Bristol, UK). Fetal bovine serum (FBS) was purchased from Gibco (Waltham, MA, USA) and penicillin, streptomycin, L-glutamine from Dutscher (Brumath, France). MEM non-essential amino acid and Cellytic™ buffer were supplied by Sigma (St. Louis, MO, USA). Minimum Essential Medium Eagle was obtained from Eurobio (Montpellier, France).

### 2.2. Cell Culture β

Human colon fibroblast cell line CCD-18Co was purchased from ATCC (Manassas, VA, USA) and used between passages 4 to 10. Cells were cultured as they were previously [1]. Human intestinal epithelial cell (IEC) lines HT-29, HCT-8 and Caco-2 were also used.

### 2.3. Cell Treatment

CCD-18Co cells were first deprived from FBS 24 h before cell induction. CCD-18Co cells were incubated with 10 ng/mL^−1^ of TGF-β1 with an increasing concentration of FICZ (10 to 1000 nM), L-kynurenine (0.1 to 10 μM), ITE (1- to 100 μM) or curcumin (5 to 20 μM) for 24 h (Figure 1). Cell supernatants were stored at −80 °C and cells were, respectively, lysed in Cellytic™ buffer or Trizol (Invitrogen, Waltham, MA, USA) for protein or RNA analysis. Each experiment was performed in duplicate at least 4 times. IEC cell lines were first FBS-deprived and induced with 10 ng/mL of TGF-β1 for 24, 48 and 56 h.

### 2.4. RT-qPCR Analysis of Cell Gene Expression

RT-qPCR was performed for alpha-SMA (α-SMA), connective tissue growth factor (CTGF), collagen 1 (COL1A1) and collagen 3 (COL3A1), AhR and ANRT as previously [1]. GAPDH was used as the endogenous reference gene (ThermoFisher, Waltham, MA, USA).

### 2.5. ECM-Associated Protein α-SMA Expression by Western Blot

25 µg of protein were separated on 4–20% gradient polyacrylamide gel (Bio-Rad, Hercules, CA, USA) by the SDS-PAGE system and then transferred to a nitrocellulose membrane. These membranes were then blocked for 1 h at room temperature with 5% of bovine serum albumin (Eurobio, Montpellier, France) in Tris-buffered saline (10 mM Tris, pH = 8; 150 mM NaCl) and 0.05% Tween 20. Membranes were then incubated overnight at 4 °C with primary antibodies: α-SMA (A5228, dilution: 1/5000, Sigma, St. Louis, MO, USA), GAPDH (SAB2500541, 1/5000, Sigma, St. Louis, MO, USA). After 3 TBST washes of 5 min each, membranes were incubated in appropriate secondary antibodies (1/5000, Dako, Produktionsvej, Denmark) 1 h at room temperature. Immunocomplexes were revealed by a chemiluminescence detection system (GE Healthcare, Chicago, IL, USA). Proteins bands were scanned (ImageScanner III; GE Healthcare, Chicago, IL, USA) and analyzed.

### 2.6. Statistical Analysis

All data were expressed as the mean ± standard error mean and were analysed using Graphpad Prism version 6.0 (Graphpad Software, La Jolla, CA, USA). Differences between two groups were assessed by the parametric Student’s *t* test or the non-parametric Mann–Whitney test, and one-way ANOVA followed by the Tukey post-test or Bonferroni post-tests were used for more than two groups. Differences were considered statically significant at *p* < 0.05.

## 3. Results

To determine whether tryptophan metabolites or curcumin have antifibrotic effects in vitro, human colonic myofibroblasts (CCD-18co) were firstly stimulated with TGF-β to induce a fibrotic phenotype. TGF-β increased expression of profibrotic genes including COL1A1, COL3A1 and CTGF (Figure 1B–D). To confirm our findings on a second intestinal cell line, we investigated the effect of TGF on three intestinal epithelial cell lines HT-29, HCT-8 and Caco-2 cells for 24, 48 and 56 h and none of them expressed α-SMA (Appendix A). In Caco-2 cells, TGF-β had no effect on the expression of profibrotic genes including COL1A1 and CTGF (Appendix A).

We then investigated whether natural agonists of AhR were able to modulate ECM-associated proteins. We found that FICZ from 10 to 1000 nM did not repress TGF-β induction of α-SMA protein expression (Figure 2A), nor repressed the expression of fibrotic genes CTGF, COL1A1 and COL3A1 (Figure 2B–D). Upon stimulation by a ligand, AhR translocates into the nucleus and the complex heterodimerizes with its partner ARNT (AhR Nuclear Translocator). It’s the reason why we determined mRNA levels for AhR and ANRT.

FICZ from 10 to 1000 nM had no effect on AhR (Figure 2E) while FICZ at 100 nM increased mRNA levels for ANRT (Figure 2F).

L-Kynurenine from 0.1 to 10 μM had no significant effect on neither α-SMA (Figure 3A), nor the expression of fibrotic genes CTGF, COL1A1 and COL3A1 (Figure 3B–D), nor AhR or ANRT mRNA levels (Figure 3E,F).

Similarly, ITE from 1 to 100 μM had no effect on studied parameters: α-SMA, CTGF, COL1A1 and COL3A1, AhR or ANRT (Figure 4). Increasing concentration of curcumin from 5 to 20 μM did not alter ECM-associated genes (Figure 5A–D), nor the expression of AhR (Figure 5E).

## 4. Discussion

Intestinal fibrosis is a common and a serious complication in IBD patients and occurs in more than one-third of IBD patients [1]. Fibrosis is a consequence of local chronic inflammation and is characterized by ECM-associated protein accumulation. These fibrogenesis processes can contribute to organ dysfunction. No specific anti-fibrotic therapy exists, and identification of novel therapeutic targets is required [11].

Myofibroblast activation is a key component of intestinal fibrogenesis. Activated myofibroblasts from various origins secrete ECM-associated proteins. Activated myofibroblasts are controlled by numerous mediators and TGF-β is the core cytokine in intestinal fibrosis. Briefly, TGF-β binding induced via SMAD signaling ECM-associated gene expression such as α-SMA, collagens and CTGF. In the present study, TGF-β treatment was able to induce ECM-associated genes such as COL1A1, COL3A1 and CTGF in colonic fibroblasts. This result is concordant with studies observed by others [7,12,13]. Monteleone et al. have incubated fibroblasts from CD patients with TGF-β1 and TNF-α and it induced ECM-associated genes such as COL1A1, COL3A1 and α-SMA transcripts and collagen secretion [7].

Current therapies have no direct anti-fibrotic effect, and novel therapies are under investigation to inhibit or reverse intestinal fibrosis. We aimed to investigate the potential of dietary AhR ligands. The AhR is a transcription factor that mediates cellular responses to various ligands. Upon ligand binding, there is a conformational change leading to AhR translocation into the nucleus and AhR with ARNT heterodimerization to induce target gene expression. AhR is also able to directly interact with the transcription factor NF-κB.

AhR is widely expressed in the gut, including epithelial or immune cells and AhR has been well documented to be involved in the regulation of intestinal homeostasis [5]. Numerous AhR ligands such as natural compounds have been described. Here, we tested tryptophan derivatives such as FICZ, ITE and kynurenine. Indeed, treatment with FICZ increased IL-22 production and therefore promoted antimicrobial molecules in antibiotics-treated mice [14]. Similarly, FICZ treatment inhibited intestinal hyperpermeability in a murine model of intestinal obstruction and improved gut barrier function [15]. Treatment with FICZ also improved DSS-induced colitis severity in Card9−/− mice and restored colon IL-22 production and antimicrobial peptides. In IBD patients, Lamas et al. have observed a decreased AhR activity in fecal samples [6]. FICZ is the most described dietary AhR ligand, and we also investigated two related molecules: ITE and kynurenine. These latter compounds induced regulatory T cells [16] and ITE improved DSS-induced colitis [16]. We also investigated the effect of curcumin, a flavonoid found in turmeric powder that elicits anti-inflammatory properties in colitis models [10,17,18].

The effects of AhR ligands on intestinal fibrosis is very limited. Treatment with FICZ from 100 to 400 nM decreased ECM-associated genes in stimulated fibroblasts from CD patients [7]. We did not observe any effect of FICZ from 10 to 1000 nM in TGF-β-stimulated fibroblasts. The discrepancy may result from the experimental procedure or fibroblast origin. Indeed, Monteleone used fibroblasts from IBD patients while we used colonic fibroblasts CCD-18Co. In addition, fibroblasts from the Monteleone study were incubated with TGF-β at 1 ng/mL while we used TGF-β at 10 ng/mL. Similarly, TGF-β at 5 ng/mL induced ECM-associated genes such as ACTA2 and COL1A1 in dermal fibroblasts while FICZ treatment at 100 nM decreased them [19]. The lack of FICZ effect in our study may be the result of more severe fibroblastic phenotype of our cells.

These observations are consistent with other studies. For example, AhR activation was able to reduce lung inflammation but not fibrosis in a murine model of chronic silica exposition [20].

We also investigated the effect of ITE from 1 to 100 μM and L-kynurenine from 0.1 to 10 μM on TGF-β-stimulated CCD-18Co. In the primary culture of human orbital fibroblasts, TGF-β at 1 ng/mL induced ECM-associated proteins such as fibronectin, collagen I and α-SMA while ITE at 1 μM reduced them [21]. In this study, ITE treatment lasted 96 h while TGF-β dose (1 ng/mL) was lower [21] compared to our experimental conditions: 24 h and TGF-β at 10 ng/mL. Treatment duration may be of special interest in the case of L-kynurenine. Indeed, Seok et al. have demonstrated that increasing the incubation temperature drastically increased AhR activity by L-kynurenine [22]. Incubation of L-kynurenine after 3 days at 37 °C increased 100-times AhR activity and they suggested that it may result from the accumulation of active L-kynurenine derivatives such as trace-extended aromatic condensation products (TEACOP) [22]. They observed that AhR activation by L-kynurenine in hepatic fibroblastic cells for 8 h had a higher biological response compared with a 4-h induction [22]. By contrast, a longer incubation time with FICZ is less efficient [22], and the authors speculated that may result from cellular turnover or FICZ metabolism [22]. These data are consistent with a study in a liver fibrosis context where ITE treatment at 1 μM for 6 days inhibited ECM-associated proteins such as α-SMA in hepatic stellate cells [23].

In addition to tryptophan derivatives, we also investigated the potential of curcumin to inhibit ECM-associated proteins in TGF-β-stimulated CCD-18Co. We chose curcumin because it demonstrated some potential interest in clinical practice in UC patients [24] and is also a very common “over the counter” therapy. In our experimental condition, curcumin treatment at 5 to 20 µM had no effect on human colonic fibroblasts. By contrast, Xu et al. used rat intestinal epithelial cells to measure the potential of curcumin to exert anti-fibrotic properties [10]. They observed that curcumin from 2.5 to 10 μM inhibit ECM-associated genes in TGF-β-stimulated IEC-6 cells and they demonstrated that this effect was mediated via the PPARγ pathway [10]. Here, we used curcumin at a similar concentration range from 5 to 20 μM and the discrepancy may thus result from cell type (epithelial versus fibroblasts) or species (rat versus human). Similarly, curcumin administration was found effective in numerous models of colitis [10,17,18], while a recent clinical trial did not observe a beneficial effect of curcumin administration to prevent CD recurrence after surgery [25]. In addition, it has been previously described that depending on the context, the ligand, and the cell type involved, the effect of AhR modulation can vary [5]. For example, curcumin treatment induced the DNA-binding capacity of AhR in human mammary carcinoma cells [26] while it exerts antagonistic properties by inhibiting AhR translocation in murine hepatoma cells [27].

## 5. Conclusions

In conclusion, TGF-β-treatment induced ECM-associated genes in human colonic fibroblasts. Next, we analyzed dietary components for their ability to inhibit intestinal fibrosis in TGF-β-stimulated colonic fibroblasts via AhR signaling and we failed to identify any of them in our experimental conditions (graphical abstract). We thus speculated that this lacking effect may be due to (i) incubation time, (ii) cell type, (iii) fibroblastic process step or (iv) choice of AhR ligands. Intestinal fibrosis is a challenge by the lack of drugs directly targeting factors involved in fibrogenesis such as myofibroblast activation. Novel therapeutics are thus eagerly awaited to inhibit or reverse intestinal fibrosis.

## Figures and Tables

**Figure 1 nutrients-14-03253-f001:**
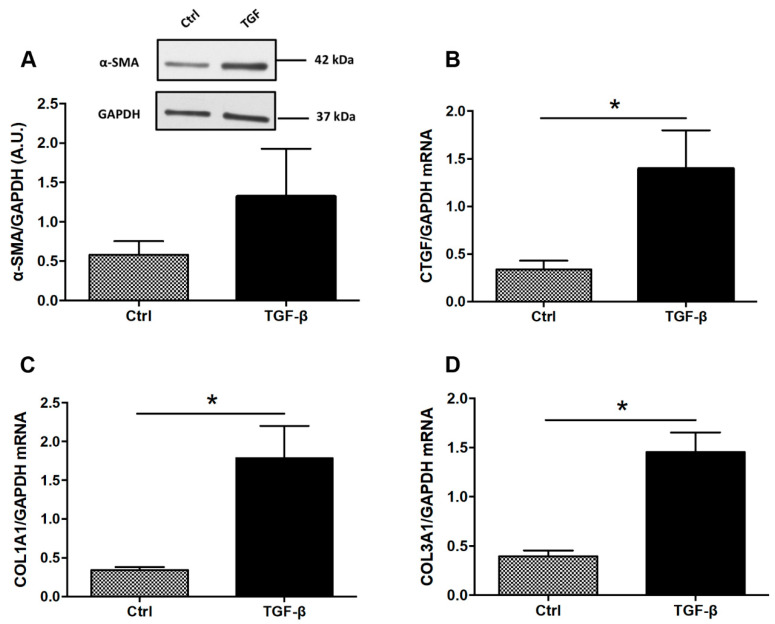
**Expression of ECM-associated proteins in TGF-β-stimulated colonic fibroblasts CCD-18Co**. CCD-18Co cells were incubated with or without TGF-β (10 ng/mL) for 24 h. α-SMA protein level (**A**) was studied by Western Blot. GAPDH expression served as the loading control for the amount of protein. mRNA levels of CTGF (**B**), COL1A1 (**C**) and COL3A1 (**D**) were studied by RT-qPCR. * means *p* < 0.05 vs. Ctrl. Results are expressed as mean ± SEM of four independent experiments.

**Figure 2 nutrients-14-03253-f002:**
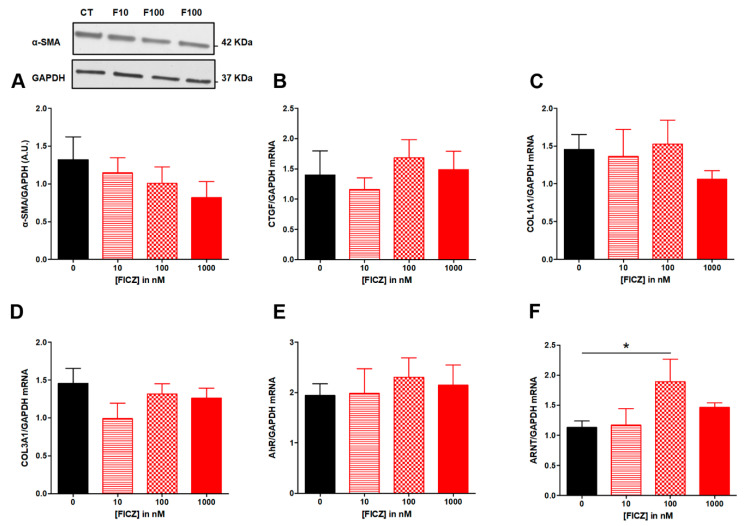
**Effect of increasing concentrations of FICZ from 10 to 100 nM on fibrosis markers in TGF-β-stimulated colonic fibroblasts CCD-18Co.** CCD-18Co cells were simultaneously stimulated with TGF-β (10 ng/mL) and FICZ (0 (CT); 10 (F10); 100 (F100); 1000 (F1000) nM) for 24 h. α-SMA protein level (**A**) was studied by Western Blot. GAPDH expression serves as the loading control for the amount of protein. mRNA levels of CTGF (**B**), COL1A1 (**C**), COL3A1 (**D**) AhR (**E**) and ARNT (**F**) were studied by RT-qPCR. * means *p* < 0.05. Results are expressed as mean ± SEM of four independent experiments.

**Figure 3 nutrients-14-03253-f003:**
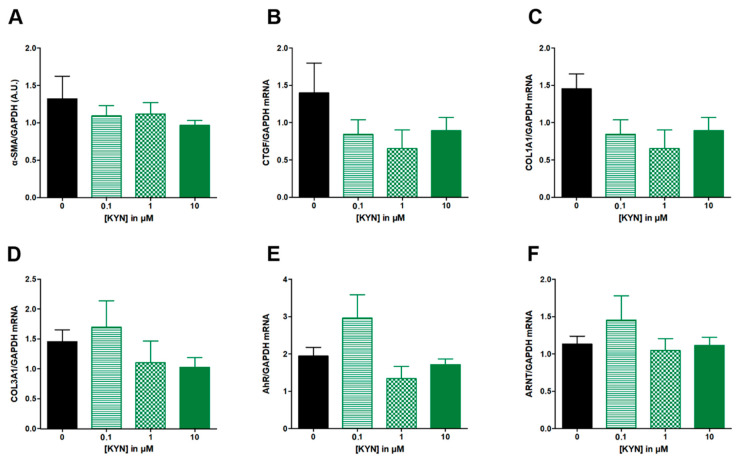
**Effect of increasing concentrations of KYN from 0.1 to 10 µM on fibrosis markers in TGF-β-stimulated colonic fibroblasts CCD-18Co.** CCD-18Co cells were simultaneously stimulated with TGF-β (10 ng/mL) and KYN (0; 0.1; 1; 10 µM) for 24 h. α-SMA protein level (**A**) was studied by Western Blot and GAGDH expression serves as the loading control for the amount of protein. mRNA levels of CTGF (**B**), COL1A1 (**C**), COL3A1 (**D**), AhR (**E**) and ARNT (**F**) were studied by RT-qPCR. Results are expressed as mean ± SEM of four independent experiments.

**Figure 4 nutrients-14-03253-f004:**
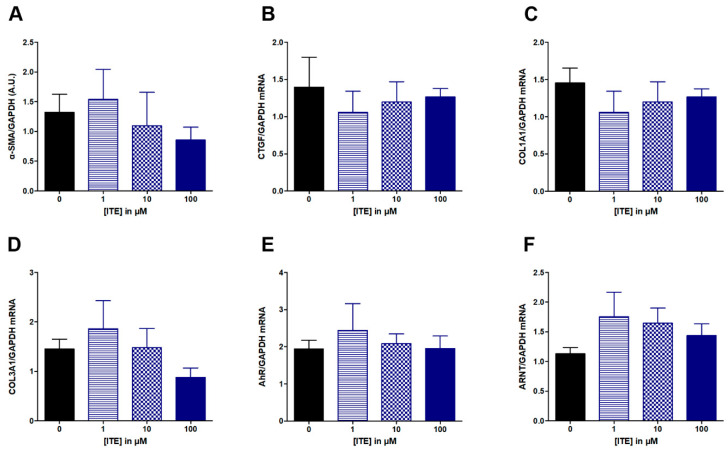
**Effect of increasing concentration of ITE from 1 to 100 µM on fibrosis markers in TGF-β-stimulated colonic fibroblasts CCD-18Co.** CCD-18Co cells were simultaneously stimulated with TGF-β (10 ng/mL) and ITE (0; 1; 10; 100 µM) for 24 h. α-SMA protein level (**A**) was studied by Western Blot and GAPDH expression serves as the loading control for the amount of protein. mRNA levels of CTGF (**B**), COL1A1 (**C**), COL3A1 (**D**), AhR (**E**) and ATNT (**F**) were studied by RT-qPCR. Results are expressed as mean ± SEM of four independent experiments.

**Figure 5 nutrients-14-03253-f005:**
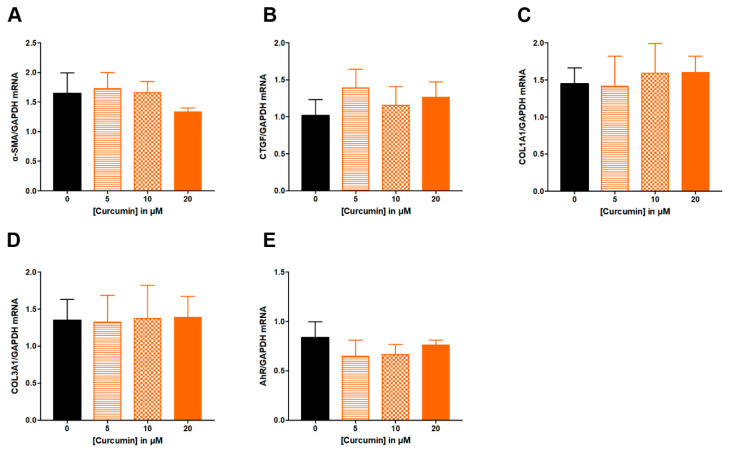
**Effect of increasing concentrations of curcumin from 5 to 20 µM on fibrosis markers in TGF-β-stimulated colonic fibroblasts CCD-18Co.** CCD-18Co cells were simultaneously stimulated with TGF-β (10 ng/mL) and curcumin (0; 5; 10; 20 µM) for 24 h. mRNA levels of α-SMA (**A**), CTGF (**B**), COL1A1 (**C**), COL3A1 (**D**) and AhR (**E**) were studied by RT-qPCR. Results are expressed as mean ± SEM of four independent experiments.

## Data Availability

Not applicable.

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
