# Peer review of "Dietary AhR Ligands Have No Anti-Fibrotic Properties in TGF-β1-Stimulated Human Colonic Fibroblasts"

_nutrients, 2022, doi:10.3390/nu14163253_

Round 1

Reviewer 1 Report

Asma Amamou and coauthors investigated the potential of dietary Ahr ligands for their antifibrotic properties in TGF-2 β1-stimulated human colonic fibroblasts.

Actually, the AhR ligands investigated by authors are endogenous, except for curcumin.  Precisely, FICZ, ITE and kynurenine are tryptophan derivatives from dietary Ahr ligands. They usually improve gut barrier function and restore colon IL-22 production and antimicrobial peptides.

While curcumin is a true dietary ligand with anti-inflammatory properties.

AhR ligands modulate the development, function, production, and maintenance of several key mucosal immune cells and mediators. Ahr transcription factor are associated with better prognosis in intestinal fibrosis.

It is known that in the gastrointestinal tract AhR ligands influence several pathways, including epithelial cell renewal and turn over, and modulate and maintain immune responses, contributing to pathogen eradication.

However, the exact molecular mechanisms are not well known.

The authors studied for the first time if AhR ligands have anti-fibrotic effect in CCD-18co human normal colon fibroblast cell line but  they failed to identify any of (published) effects in their experiments .Therefore they speculated that the discrepancy respect other papers is due to incubation time and choice of Ahr ligands.

It is true that the origin and structure of AhR ligands are diverse and their effects are not similar, but the authors should consider an additional cell line other than the stimulated CCD-18co human normal colon fibroblast cell line to evaluate the anti-fibrotic effect of tryptophan derivatives AhR ligands and curcumin.

Finally, to conclude that the effects of Ahr ligands on intestinal fibrosis is very limited it would be advisable to use a control in which the anti-inflammatory effect of the activation of the AhR pathwy with the ligands is clearly seen.

Major revision:

The authors should confirm their results in an additional cell line.

Minor revision:

1.     Figure 1 adds nothing to the materials and methods, it should be eliminated

2.     The legends of the figures cannot be read, they should be enlarged

Reviewer 2 Report

In the manuscript "Dietary AhR ligands have no anti-fibrotic properties in TGF-β1-stimulated human colonic fibroblasts" by Amamou et al, the authors describe effect of AhR ligands on TGFb1 stimulated human colonic fibroblast cell line (CCD-18Co).  Even though the manuscript investigate very interesting and clinically relevant topic, the results are negative.

Reviewer 3 Report

The authors of “Dietary AhR ligands have no anti-fibrotic properties in TGF-β1-stimulated human colonic fibroblasts.” investigated the effect of increased concentrations of dietary ligands of AhR such as FICZ, ITE, L-kynurenine and curcumin on markers of fibrosis.

The results do provide important insight into ineffectiveness of the tested compounds on the TGF-β1 induced fibrosis, however the data is not enough to hold as a publication alone. As the authors suggest in the conclusion, the lack of response can be due to incubation time or cell type used.

It would be more valuable to test with longer incubation times and other intestinal cell lines.

Most experiments were performed 4 times. Is each experiment in triplicates?

Round 2

Reviewer 1 Report

In the point by point reply in response to my suggestions the authors wrote that they “tested three intestinal epithelial cell lines (HT-29, HCT-8 and Caco-2) in response to TGF-beta and none of them over-expressed ECM-associated proteins in response to TGF-beta. These three cell lines have been tested at three incubation times (24h, 48h, 56h)”.

They have to insert these sentences in material and methods, and in results. In case, they have to prepare a supplemental figure with (negative) results from the three intestinal epithelial cell lines.

Author Response

As suggested by the reviewer, we now added this point in the material and methods and result sections. We also provide a supplementary Figure.

Reviewer 3 Report

Thanks for corrections and clarifications

Author Response

No requested modifications. We now provide a supplementary figure with the data from the three tested intestinal epithelial cell lines.